Blockchain for genomics and healthcare: a literature review, current status, classification and open issues

Adanur Dedeturk Beyhan beyhan.adanur@agu.edu.tr
Soran Ahmet
Bakir-Gungor Burcu burcub@gatech.edu
Department of Computer Engineering, Abdullah Gul University , Kayseri , Turkey
Chai Tianfeng
Electronic publication date: 2021 Sep 30
Publication date: 2021
Volume: 9
Electronic Location ID: e12130
Received 2021 Feb 17; Accepted 2021 Aug 17
Copyright: ©2021 Adanur Dedeturk et al.
Copyright year: 2021
Copyright holder: Adanur Dedeturk et al.
License: This is an open access article distributed under the terms of the Creative Commons Attribution License, which permits unrestricted use, distribution, reproduction and adaptation in any medium and for any purpose provided that it is properly attributed. For attribution, the original author(s), title, publication source (PeerJ) and either DOI or URL of the article must be cited.
License URL: https://creativecommons.org/licenses/by/4.0/

Keywords: Blockchain, DNA sequencing, Genomic data privacy, Data sharing, Smart contracts, Homomorphic encryption

Funding: The authors received no funding for this work.

==============================
The tremendous boost in the next generation sequencing technologies and in the “omics” technologies resulted in the generation of hundreds of gigabytes of data per day. Nowadays, via integrating -omics data with other data types, such as imaging and electronic health record (EHR) data, panomics studies attempt to identify novel and potentially actionable biomarkers for personalized medicine applications. In this respect, for the accurate analysis of -omics data and EHR, there is a need to establish secure and robust pipelines that take the ethical aspects into consideration, regulate privacy and ownership issues, and data sharing. These days, blockchain technology has picked up significant attention in diverse fields, including genomics, since it offers a new solution for these problems from a different perspective. Blockchain is an immutable transaction ledger, which offers secure and distributed system without a central authority. Within the system, each transaction can be expressed with cryptographically signed blocks, and the verification of transactions is performed by the users of the network. In this review, firstly, we aim to highlight the challenges of EHR and genomic data sharing. Secondly, we attempt to answer “Why” or “Why not” the blockchain technology is suitable for genomics and healthcare applications in detail. Thirdly, we elucidate the general blockchain structure based on the Ethereum, which is a more suitable technology for the genomic data sharing platforms. Fourthly, we review current blockchain-based EHR and genomic data sharing platforms, evaluate the advantages and disadvantages of these applications, and classify these applications using different metrics. Finally, we conclude by discussing the open issues and introducing our suggestion on the topic. In summary, to facilitate the diagnosis, monitoring and therapy of diseases with the effective analysis of -omics data with other available data types, through this review, we put forward the possible implications of the blockchain technology to life sciences and healthcare.

Introduction

Using genomic datasets, the researchers attempted to untangle the molecular mechanisms of human diseases, contributing to identifying disease-specific mutations. The advancements in the next-generation sequencing technologies resulted in the generation of hundreds of gigabytes in a single run, and up to two billion human genomes are expected to be sequenced in the next ten years (Diniz & Canduri, 2017). Since the high-throughput technologies generate genomics data in high quantities, the management, analysis, and storage of these datasets require specific infrastructures and pipelines. In order to illuminate our genome and to uncover the hidden mysteries of complex diseases further, new analysis techniques and strategies need to be implemented. However, it is tough to make new studies because of the limitations on data management. Data collection, data sharing, analysis cost, data ownership, privacy, and security are all major concerns for today’s healthcare data management systems. Reaching data is the most critical and valuable thing in this field. Due to a single point of failure problem, Data Cloud Architecture has been developed for storing the data secure and safe. However, these architectures are not adequate to apply analysis tools and focus on storing data systematically. In addition to data and cloud infrastructure, adding typical software applications and analyzing tools have emerged a new term: Data Commons. However, this is still not enough since the bioinformatics research field requires several people from various backgrounds. Recently, researchers focus on creating a complete Data Ecosystem that can easily feed itself with APIs and interconnected Data Commons (Grossman, 2018). Here the scalability and flexibility could be the biggest problem for data management. Blockchain, on the other hand, is a decentralized technology that could completely change the way data handling in the healthcare industry (Srivastava, Parizi & Dehghantanha, 2020). Blockchain offers an adaptive, dynamic, distributed, flexible, and secure platform that can also support numerous analyzing tools, but it has some pitfalls, like inefficient data storage capacity and consensus cost, because of its peer-to-peer backbone. In this study, we examine the problems in the field of healthcare and the fundamentals of blockchain technology for giving possible solutions to fix these problems. We present and discuss recent studies to demonstrate how blockchain technology can be used in a variety of healthcare applications, and the blockchain evaluation timeline is discussed in detail. We believe that blockchain technology has enormous potential to be a valuable cornerstone while building a Data Ecosystem in healthcare applications. Before going on to the description of the studies, we examined the introduction section in four subsections in order to explain the required information for the background in detail and to show the contributions of this article.

Challenges in genomics

Although the cost of sequencing reduces much more than the estimated levels using Moore’s law, still one of the challenges in genomics is its affordability (Tibbetts, 2018). A recent study conducted by Schwarze et al. (2019) analyzed the total cost of using genome sequencing in routine clinical care. It is reported that the cost is underestimated if only sequencing costs are considered, and it likely surpasses $1000/genome in a single laboratory. This situation complicates participation in a specific sequencing project because, in classical systems, the transactions are carried out with the help of a middleman, as opposed to a direct relationship between the data owner and the buyer. Another challenge in genomics field is data management, which deals with the following four critical issues: (i) collection, (ii) sharing, (iii) ownership, and (iv) storage. (Steneck, 2017) stated that the person who conducts the research should own the data and be aware of its responsibilities. Hence, researchers put lots of effort into data collection. As a result, numerous data collection methods are proposed with the following four essential considerations: (i) appropriate methods, (ii) attention to detail, (iii) authorization, and (iv) recording. In order to conduct high-quality research, special attention should be given to details and the results should be accurately recorded and interpreted. In the first phase of the data collection, a person or group responsible for the research must be authorized. All related permissions must be taken, and all requirements must be fulfilled. Genomic data sharing is a very sensitive issue since genomic data carry private information about an individual’s past, present, and future. One potential misuse of genomic data could be the use of synthesized genomic data in crimes. Another potential misuse of genomic data could be the development of harmful medicines (Humer & Finkle, 2014). People want to ensure that their personal data is kept at high-level protection and privacy. In this respect, some other fears include not being able to control data access permissions and whether full anonymity could be provided (Hubaux, Katzenbeisser & Malin, 2017).

Generally, genomic data are stored in databases by governments. Public databases only display summary data or frequency information. The amount of genomic data is expected to exceed the amount of video and content data in the next decade since the sequencing of a single human genome regularly produces ∼200 GB of data (may vary depending on the sequencing type). According to the estimates, around 2 billion human genomes will be sequenced by 2025 (Stephens et al., 2015). The investigation of huge genomic data requires many disk spaces for storage, fast transfer speed for data sharing and fast processing power since the analyzes take trillions of CPU hours (Stephens et al., 2015; Ozercan et al., 2018). While sharing genomic data offers the unique opportunity to increase our knowledge by obtaining novel information from the re-analysis of the same datasets and collective datasets, it imposes several challenges of ethical, legal and technical nature. In this respect, recently, blockchain technology has picked up significant attention in diverse fields, including genomics, since it offers a new solution for these problems from a different perspective: distributed secure and immutable system. With its potential to solve several security and agreement issues, including data sharing and secure computing in a public network, blockchain-based platforms have a snowball effect.

General structure of blockchain technology

Blockchain is an indefectible distributed ledger of transactions (Yaga et al., 2019). The best-known applications of this technology are Bitcoin (Nakamoto, 2008) and Ethereum. It builds a chain model that can be tracked transparently but could not be broken. Blockchain allows operation without having a central authority, so it prevents single-point-of-failure with a distributed timestamp mechanism. In this network, transactions can be directly and safely executed among all participants. Before the development of blockchain technology, the primary problem for digital payment systems was a double-spending problem (Kuo, Kim & Ohno-Machado, 2017). The double-spending problem is to spend the same digital currency more than one time. To prevent this issue, blockchain uses different verification mechanisms implemented via various consensus algorithms. Consensus algorithms are agreements for the validation of transactions among a group of individuals. At the end of the verification process, the majority voting makes the decision (Mingxiao et al., 2017). Consensus-based decision-making is more effective than single authority-based decision-making since the whole group is taking action on the decision. In this way, the power distribution in a group is equalized.

As summarized in Table 1, the selection of consensus algorithms differs based on the blockchain application. In blockchain structure, the transactions are processed on the blocks with their hash values instead of using the original data. A hash function takes any string as an input and generates a fixed-size output. Since cryptographic hash functions satisfy the following three properties, i.e., collision freeness, hiding-binding and puzzle-friendliness, it is tough to find two different messages with the same hash output. It is not guaranteed to be impossible, but the probability of such an event is very low (Preneel, Govaerts & Vandewalle, 1993).

Table 1 Different types of consensus algorithms (Anwar, 2018).

Consensus algorithms	Explanations	
PoW	When a user initiates a transaction, miners try to solve a cryptographic problem to verify that they worked a lot.	
PoS	A user encouraged to spend more until he becomes a validator to create a block.	
PoWeigth	Similar to PoS but the difference is that it depends on various other factors called weights.	
PoB	Users send the coins back into their wallet that they can’t recover from will get rewards based on the amount.	
PoC	Using this protocol you can utilize the capacity of user’s hard drive.	
DPoS	Same as PoS but users with more coins will get to vote and elect witnesses.	
DBFT	Focuses on a gamified way of a block verification among the professional node controlles.	
PBFT	Byzantine used a particular sequence to keep the rouge users at bay.	

In the blockchain network, each block holds the hash value of the previous block like a chain structure and this structure creates immutability. Also, in blockchain systems, all users have public and private keys instead of using their real identities. While the public keys are the keys that everyone knows, private keys are unique for each user and are used to sign the transactions. Hence, the very first version of the blockchain, the bitcoin network, is called pseudo-anonymous. Miners pick up a set of waiting transactions from the mem-pool and try to create a proper block that provides a given cryptographic puzzle solution. Problems or puzzles could differ according to different blockchain protocols, but the common thing is the usage of a hash function. In bitcoin, the hash puzzle is to find a hash value that starts with a particular number of zeros, called proof-of-work. When the desired output is obtained by one of the miners, the generated block is broadcasted to everyone in the network. At the same time, to keep the miners in the system, the miner who finds the new block gets a reward as an incentive (Angraal, Krumholz & Schulz, 2017).

Ethereum-smart contracts

Ethereum (https://ethereum.org/) is a decentralized platform based on blockchain technology. Among other blockchain technologies, the benefits of Ethereum are widely recognized in areas like global public health, pharmacology and medicine (Agbo, Mahmoud & Eklund, 2019). Since Ethereum runs with smart contracts, it could be used in genomics and healthcare applications (Kuo, Rojas & Ohno-Machado, 2019). A Smart Contract is an option that creates an agreement between parties with a written code into the blockchain (Macrinici, Cartofeanu & Gao, 2018; Wang et al., 2019). While participating individuals are anonymous, the contract is on the public ledger. When a triggering event happens (for example, when a genomic-data request occurs or when a patient wants to be examined), the contract executes itself according to the coded terms. Smart contracts consist of two types of accounts, (i) owned account and (ii) contract account, as shown in Fig. 1. Ethereum has a virtual machine called Ethereum Virtual Machine (EVM). The use of EVM is required to participate in the Ethereum system. Also, the users should spend EVM GAS, which is considered as a fuel of contract execution to deal with Denial-of-Service (DoS) attacks. When the contract executes, the gas price is paid. Another feature of Ethereum is its DApps. Ethereum DApps is a service that could be enabled with direct communication between users and providers. It is interfaced with a user via HTML/Javascript web application using a Javascript API for communicating with blockchain. It also has its own suite contract on the blockchain. In the future, DApps are likely to be listed and distributed in DAppstores integrated with DApps browsers. In conclusion, the benefits of using a smart contract are as following: (i) data storage and protection, (ii) managing relationships and agreements, (iii) providing functions to other contracts, and (iv) sophisticated authentication. With DApps, blockchain network can run any software tools and update the list of supported tools in a decentralize way, and this is suitable for creating a Data Ecosystem.

Figure 1 An example of account types of ethereum.

Why/why not blockchain technology is suitable for bioinformatics and healthcare applications?

The blockchain can reduce the analysis cost of genomics and healthcare applications. Contrary to the functioning of existing systems, if data owners can contact data buyers directly without an intermediary company, both the analysis costs decrease and the data owners can earn an income. Those statements are also valid for electronic health care applications. With blockchain systems, transactions are completed fast and more efficiently than conventional processes. In the central systems, recorded data can be manipulated; however, in blockchain, the immutable ledger is generated with the participation of all the users. Thus, transaction histories became more transparent and traceable with blockchain since it is a distributed system. It also prevents fraud and unauthorized activity, much more than other systems. Transparency and traceability are obligatory for the verification mechanism (Kumar et al., 2018; Stagnaro, 2017).

If more data can be safely shared, more improvements in the field of healthcare and genomics can be made. Personal data security and privacy are very sensitive. Although any system cannot provide full anonymity (Wust & Gervais, 2018), Bitcoin provides pseudo-anonymity. For genomics and healthcare, another sensitive issue is that people do not want to share the original data directly. For instance, by examining the whole genome data of an individual, the person can be detected, or even information about his/her ancestors can be obtained. In blockchain systems, transactions are stored with their hash values instead of the original data. Also, individuals only share metadata, which includes general information about the data. Blockchain allows individuals to edit their own data access permissions (Mackey et al., 2019). With smart contracts, people can edit with whom to share their data. Additionally, some encryption methods can be applied, like homomorphic encryption (HME) (Acar et al., 2018). It allows computation on encrypted data without decryption. There are three types of HME, i.e., partially, somewhat, and fully. Fully HME has no application in practice (Frederick, 2015). Generally, partially-HMEs are preferred by genomics applications of blockchain (Moore et al., 2014). The blockchain is known as a secure system because it uses consensus algorithms for resisting the formation of an unsafe environment (Yang & Yang, 2017). The current healthcare management system needs interoperability for all country individuals because there is no universal standard for it. By specifying what kind of data, size, and format to use, the information is saved in blocks; hence, the blockchain guarantees to engage EHR interoperability. As a limitation, full anonymity can be provided neither by current nor by blockchain-based systems for genomics and healthcare until people will obtain their own analysis results with a portable device (Krawiec et al., 2016; Gordon & Catalini, 2018).

Rationale of the review and intended audience

We believe that EHR and genomic data sharing will become widespread by taking advantage of the innovations brought by blockchain technology. Public health will be positively affected by this development. In line with this purpose, several projects have been developed during the last five years. The main goal of this article is to review these existing blockchain-based genomic data sharing and EHR sharing platforms. Another aim of this review is to attract the attention of scientists to this field and boost the informatics/bioinformatics community to develop new approaches for solving the problems on these issues. Recently, some review papers are published on similar topics (Casino, Dasaklis & Patsakis, 2019; Drosatos & Kaldoudi, 2019; Abu-elezz et al., 2020; Hasselgren, et al., 2020; Ramachandran et al., 2020; Ahmad et al., 2021; Sookhak et al., 2021; Yaqoob et al., 2021; Omar et al., 2021). To the best of our knowledge, in literature, there is no survey that: (i) reduces the subject titled “blockchain technology in healthcare” from general to specific, (ii) elaborates on EHR sharing and genomic data sharing platforms, (iii) explains their functioning mechanisms, (iv) demonstrates their advantages-disadvantages, (v) classifies and compares current platforms, (vi) demonstrates how blockchain-based projects have changed between 2016 and 2021, (vii) aims to show the timeline of blockchain use in healthcare and to discuss future potentials. For these reasons, we believe that for our target audience, this study is a more selective review in the above-mentioned topics than the current review articles.

Survey Methodology

In this review article, we examine EHR sharing and genomic data sharing platforms. In this respect, using Web of Science, Scopus and Google Scholar, we searched the terms “Blockchain-based EHR sharing”, “Blockchain-based genomic data sharing”, “Blockchain-based applications”, and “Blockchain technology in healthcare”. We would also like to point out that blockchain-based platforms generally have white papers and unique websites instead of academic papers between 2016 and 2018. For this reason, we explained some of the projects using their own documents. Also, it should be mentioned that some of these projects do not exist or are not supported anymore. In the period from 2019 to the present day, academic studies are carried out on this subject. We chose the articles from those precisely related to the specified titles and those who are designing a new platform. As a result of this selection mechanism, we came across 12 project-based studies between 2016 and 2018, and 9 research papers in the period from 2019 to the present day. These 12 projects are Nebula Genomics, Zenome, Genecoin, Gene-Chain, DNATIX, Medrec, IRYO, Coral Health, Patientory, Medicalchain, GemOS and e-Estonia. While 5 of these projects are genomic data sharing platforms, 7 of them are EHR sharing platforms. The other 9 papers are also related to EHR sharing platforms. But to demonstrate the evolution of blockchain use in healthcare, we wanted to examine the projects in three-part as the proof of concept era, the blockchain development era and the blockchain as a platform (BaaP) era in EHR sharing.

Blockchain-Based Systems

The aim of this study is (i) to classify and specify the advantages and shortcomings of current blockchain-based bioinformatics and healthcare applications, and (ii) to demonstrate the evolution of blockchain use in healthcare over time, as shown in Fig. 2. For these purposes, we came across 12 project-based studies in the proof of concept era. We divided the systems into two main topics: (i) genomic data sharing and (ii) EHR sharing. Among the examined twelve projects, Nebula Genomics, Zenome, Genecoin, Gene-Chain and DNATIX, have been developed for genomic data sharing, but Medrec, Coral Health, IRYO, Patientory, Medicalchain, GemOS, and e-Estonia have been developed for EHR sharing. They mainly have white papers and unique websites instead of academic papers. For this reason, we explained some of the projects using their own documents. The main features of platforms developed in the proof of concept era are summarized in Table 2. This table shows that (i) the platforms that the project runs on, (ii) the information about the project and the establisher, (iii) the focus area of the project, and (iv) the main features of the project.

Figure 2 Timeline of blockchain use in healthcare.

Table 2 Comprehensive evaluation of projects in proof of concept era.

Project	Platform	Country	Company and year of establishement	The focused area	Register
kit	Mobile
application	Patient
monitoring	Disease prediction
(AI)	
Nebula Genomics
(Grishin et al., 2018)	Ethereum	USA	Nebula Genomics
2016	Genomic and Phenotyping Data Sharing	+	No exact information	-	-	
Zenome (ZNA)
(Kulemin, Popov & Gorbachev, 2017)	Ethereum	Russia	Zenome
2017	Genomic Data Sharing	-	No exact information	-	+	
Genecoin
(Schorchit et al., 2018)	Ethereum	Brasil	Genecoin
2017	Genomic Data Sharing	+	+	-	-	
Genechain (DNA)
(Encrypgen, 2017)	Hyperledger (https://www.hyperledger.org/)	USA	EncrypGen
2016	Genomic Data Sharing	+	-	-	-	
DNATIX
(DNAtix)
(DNATIX, 2018)	Ethereum	Israel	DNAtix
2014	Genomic Data Sharing	-	No exact
information	-	-	
Medrec
(DNATIX, 2018)	Ethereum	USA	MIT Media Lab
2016	EHR Sharing	-	+	+	-	
Iryo
(IRYO)
 (IRYO, 2017)	EOS	Slovenia	IRYO
2017	EHR Sharing	-	+	+	+	
Coral Health
(Park et al., 2017)	Ethereum	USA	Coral Health
2017	EHR and Genetic Test Results Sharing for Personalized Medicine	-	+	+	-	
Patientory
(PTOY)
(Mcfarlane et al., 2017)	Ethereum	USA	Patientory
2015	EHR Sharing	-	+	+	-	
MedicalChain
(MTN)
(Medicalchain, 2018)	Hyperledger Ethereum	UK	Medicalchain
2017	EHR Sharing	-	+	+	-	
GemOS
(Kannan & Smith, 2016)	Hyperledger Ethereum	USA	GemOS
2016	EHR Sharing for Personalized Medicine	-	+	No exact Information	No exact Information	
e-Estonia
(e Estonia, 2012)	KSI	Estonia	Guardtime
2009	EHR Sharing and Electonic Prescription	-	+	+	-	

From 2019 to the present day, academic studies are carried out on blockchain-based EHR management subjects instead of genomic data sharing. Therefore, we examined EHR sharing systems in the different time frames to demonstrate the evolution of blockchain technology in EHR management. In this direction, EHR sharing systems between 2016–2018 are examined under the title of the proof of concept era in EHR sharing, between 2019–2020 studies are considered as the blockchain development era in EHR sharing, and the studies after 2021 are discussed in the blockchain as a platform era. The main features of 9 studies developed between 2019 and the present day are shown in Table 3. This table shows (i) the blockchain type, platform and consensus mechanism, (ii) data-keeping strategy, (iii) performance evaluation, and (iv) the supports such as disease prediction or patient monitoring. Proposed platforms in this time window are as follows; Liu et al. (2019), IBM’s Medical-Blockchain, Al Omar et al. (2019), Tanwar, Parekh & Evans (2020), Niu et al. (2020), Veeramakali et al. (2021), Połap, Srivastava & Yu (2021), Chen et al. (2021) and Arul et al. (2021).

Table 3 Comprehensive evaluation of  projects in blockchain development era and blockchain as a platform era.

Studies	Blockchain type	Platform	Consensus	EHR storage	Performance
evaluation	Patient
monitoring	Disease prediction (AI)	
Liu et al. (2019)	Private	N/A	DpoS	On chain: Private	Based on EHR sharing-Blockchain	–	-	
IBM’s Medical-Blockchain	Private	Hyperledger	N/A	Off chain: Cloud	Based on EHR sharing-Blockchain	–	-	
Al Omar et al. (2019)	Private	Ethereum	N/A	On chain: Permissioned	Based on EHR sharing-Blockchain	–	-	
Tanwar, Parekh & Evans (2020)	Private	Hyperledger	BET	On chain: Private	Based on EHR sharing-Blockchain	-	-	
Niu et al. (2020)	Private	Ethereum	PoW	Off chain: Local cloud	Based on attribute-Based Encryption	-	-	
Veeramakali et al. (2021)	Private	N/A	N/A	On chain: Private	Based on
prediction model	+	+	
Połap, Srivastava & Yu (2021)	Private	N/A	N/A	Off chain: Database	Based on operation of learning agent	+	+	
Chen et al. (2021)	Private	N/A	N/A	Interplanetary File System	Based on
prediction model	+	+	
Arul et al. (2021)	N/A	N/A	N/A	Off chain: Database	Based on EHR sharing-Blockchain	+	+	

In this section, we examined the projects according to the roles, transactions and oracle of blockchain in detail. In the conclusion section, all fundamental features of the projects are summarized and classified according to specified metrics. Also, both common advantages-disadvantages of projects and the unique advantages-Itages of each project are discussed.

Proof of concept era

In the proof of concept era, the proposed applications require core development since the blockchain concept is new to the field. Please note that, in this era, the first target is to present a working application to prove that the blockchain platforms can be used in the healthcare management systems. Blockchain-based platforms in this section; (i) covered both genomic data sharing and EHR sharing, while in other ages, studies are done entirely on EHR sharing, and (ii) designed the detailed architecture of blockchain-based healthcare systems, while in other ages, the proposed methods include other techniques in the systems by reducing the blockchain part over time.

Genomic data sharing

Nebula genomics.

Nebula Genomics is an Ethereum-based genomic data sharing and analysis platform; and it does not have a token yet. They aimed to overcome four fundamental problems, i.e., reducing sequencing costs, data protection, data acquisition, and big genomic data. Nebula network also aims to absorb the forthcoming data explosion (Grishin et al., 2018). In the traditional model, individuals share their data with an intermediary company. When a person wants to sequence his/her genetic material, he/she should contact a company that conducts the genetic analysis. After analyzing the genomic data, companies send only analysis results instead of the whole sequencing data, and they might sell genomic information to other companies since individuals cannot take control of their data access permissions. However, in the Nebula network with blockchain, individuals can share their data directly with their own terms, and gain a profit as shown in Fig. 3A. To use this network, people need to have their sequence analysis, which can also be provided by Nebula’s sequencing facility. It is claimed that a person can sequence his/her data at lower prices in this facility. Veritas Genetics is a partner of Nebula and thus, individuals mail their sample to Veritas Genetics to get the sequences on Nebula Servers. Whole sequencing data are given to the person, and it is erased from Nebula Servers. Although it is claimed that the data is deleted, full data privacy could not be ensured. The network consists of data owner nodes, data buyer nodes, secure-compute nodes, and Nebula servers.

The working principle of the Nebula system depends on the secure multi-party computation (Choi & Butler, 2019). In the network, data buyer nodes order genomic and phenotypic data with Nebula tokens and analyze the data only on a secure compute node because shared data must be encrypted with the HME format. Secure compute nodes run the bioinformatics platform called Arvados which supports Intel SGX and partially homomorphic encryption (Sadat et al., 2018). Intel SGX is a set of instruction codes and it allows the creation of private memory regions, which are called enclaves (Drucker & Gueron, 2017). Encrypted data is transferred to a secure computing node by the data owner. In the SGX enclave, the data is decrypted, further computations are performed, results are encrypted in the SGX enclave, and forwarded to a buyer node. Thus, buyers are able to get the results without seeing the data itself that partially solves the privacy problem.

Figure 3 (A) Nebula model; (B) Medrec model.

Zenome.

Zenome, a ZNA token, is an Ethereum-based genomic ecosystem that focuses on genetic data sharing. With Zenome, data owners can share their own data, edit data access permissions, store data safely, earn money, propose recommendations for participants, perform genetic analysis and use other genetic services (Kulemin, Popov & Gorbachev, 2017). In the system, metadata is stored on a distributed network anonymously. Thanks to the data encryption, accessing to a participant’s genetic information is impossible unless the user lets others reach the original data. The system has four types of nodes, i.e., (i) Calculation and storage nodes provide storage and CPU power to get the rewards; (ii) People nodes provide genetic data and user services; (iii) Analyst nodes analyze genetic information; (iv) Service providers present a genetic service in return for payment on the platform. Players of the genetic services are scientific corporations, bioinformatics companies, medical centers, and laboratories. The Zenome is also suitable for other organisms and provides reports to the users if needed. The system uses artificial intelligence (AI) methods and all users need to use the Zenome software tool. Zenome software tool allows the users to arrange their own data privacy, such as full privacy, standard privacy, or public access options. The system can detect fake data during raw data preprocessing. Initially, the system looks for coverage according to a threshold and stores them based on fragments to deal with fake organisms. Thus, only a small amount of data is transferred through the blockchain. The rest of the data could be transferred through an encrypted communication channel.

Genecoin.

Genecoin, GEN token, is a bio-economy currency that provides sharing of genomic data securely based on the Ethereum blockchain (Schorchit et al., 2018). If one of the users can recruit another user to the global Genecoin network, the system will send incentives, up to one thousand Genecoin, without a cost. It samples an individual’s DNA and stores it in the blockchain network. The genetic material of people is spread to thousands of computers in the networks. The company sends a kit to collect a sample from the individual and then communicates with the third-party service providers to obtain the sequencing data. The system provides software, which can extract and decrypt the genome of users from the blockchain. Founders hold %10 of the tokens, and %15 of the tokens are distributed among supporters.

Gene-chain.

Gene-chain is a Hyperledger-based blockchain application that focuses on a safe, traceable, and unhackable method for transactions, consisting of genomic data (Encrypgen, 2017). It does not have a token yet. In the system, when data is uploaded, the access permissions are organized entirely by the owner. Each user has a copy of the most recent ledger; hence, the database of the platform is shared and decentralized. Also, data sharing is performed like peer-to-peer, so there is no intermediary. The data identity is pseudo-anonymous since they use public and private keys. Re-identity of them is possible, even though unlikely (Malin & Sweeney, 2004). Patients can upload data freely, but they may upload fake data. To solve this problem, the system claims that it has an ability to remove unauthorized data.

DNATIX.

DNATIX, DNAtix token, is an Ethereum-based project to provide a transparent, accessible, and secure platform for data sharing to consumers, researchers, laboratories, and clinics. It allows users to upload users’ genetic data anonymously and it performs a genetic test on these data according to the access permission conditions set by the users. In addition to other systems, one of the key improvements in this system is as follows. The participants can earn tokens via developing a new application on Next Generation Decentralized Genetic Application (GDAPPS) which has its own virtual machine like DNAtixVM and it can be run as a node. Another advantage of DNATIX is its own compressing algorithm for compressing long DNA sequences. The current algorithm of it can compress %25 of the sequence’s size. Similar to other projects, the platform uses smart contracts to store, transfer, and genomic test data. Also, the platform can revoke and update access. In order to determine the transaction fee (gas), they found a suitable number of nucleotides (DNATIX, 2018).

Electronic health records sharing systems

Medrec.

The first implementation of MedRec is announced in 2016 (Azaria et al., 2016). The current version of Medrec, MedRec 2.0, does not have a token yet and it is still under development by MIT Media Lab with GO-Ethereum and Solidity. Medrec is a blockchain-enabled Electronic Health Record sharing platform; it works safely, transparently between patients and providers; and it is scaleable. The Medrec system provides to patients transparency, quick access, and the correction of errors by the authority for the patients’ records. The system is built on smart contracts. The records are not stored directly by the Medrec; only the metadata of the information is stored. To provide system security and patient privacy, the blockchain is maintained by the providers and a selected group performs the consensus. In the system, there are 3 types of contracts; registrar contract, patient-provider relationship contract, and the summary contract. Registrar contract includes participant IDs for their Ethereum identity. Patients can update their information and this operation is added to the registrar contract only if it is confirmed by the patient. Patient-provider relationship contract connects patient node and provider node and contains their relationship. Finally, summary-contract consists of different relationships of the participants in the system. It includes a reference list to the patient-provider relationship contracts. Each relationship has a status value, which shows when the relationships were established and what the permissions are (Lipman et al., 2017). When a service provider wants to update a health record of a patient, it creates a request with the patient’s ID, as shown in Fig. 3. (b). The request is processed on the blockchain network, and the contract is examined. If the Medrec system gets all confirmations, an update is carried out successfully and the patient is informed about this update. When a patient requests access to his/her health records, the request is sent to the Database Gatekeeper of the provider. The Database Gatekeeper provides a tool for accessing the participants’ local databases. Then, the request is processed on the blockchain and the summary contracts and the patient-provider contracts are examined by the system. If the system verifies the contract, EHR is shared with the patient. For privacy, Medrec uses a system of delegated contracts where each provider creates a different Ethereum identity for each new patient-provider relationship. In terms of scalability, the platform does not have an exact solution since scalability is one of the critical challenges for any blockchain system.

IRYO.

It, the IRYO token, is the EOS-based health record sharing and storage platform (IRYO, 2017). It allows the preparation of permission controls and performs AI-based research. Several AI methods require to access patient data in order to formulate and test a new algorithm for early-stage detection of diseases and developing new treatments. These systems can be based on OpenEHR (Atalag et al., 2013) data modeling and sharing. Although the OpenEHR community continues to collect data for 15 years, they cannot exchange the data globally and they cannot collect data from more extensive areas. The IRYO is the global storage of EHR and it aims to share the archetypes data such as blood pressure with mobile devices for predicting a condition. Instead of holding the data in one place, the users encrypt their data on their mobile devices with a public key and hence the attacks are prevented. The private key decryption is performed on the devices of the patients. When someone wants to access patient data, this request must be approved by the patient who is using the IryoEHR application. Then, the doctors can obtain a re-encryption key to access the data. Encrypted health records are stored on three types of nodes. The first copy is kept on the IRYO cloud node, the second copy is stored on the home clinic storage node, and the last one is stored on the end-user device nodes.

Figure S1 shows the architecture of the IRYO nodes. To keep the data storage system secure from possible risks, IRYO keeps at least one encrypted alive copy. In this way, it can guarantee protection for any kind of health data. To use the stored data in AI methods, first, the research institution must be verified by IRYO. Researchers should use the IRYO research software tool to make a new request. If the submitted query of a researcher matches a patient’s criteria, the patient gets a notification on his phone. As soon as the patient accepts the process, the data is shared with the research institution, and the patient earns some tokens. Once the researcher’s analysis is finished, the patient gets a notification on his phone related to the analysis results. In the system, researchers can study both anonymous and pseudo-anonymous personal data. The IRYO network is a public blockchain. All storage nodes supply cryptographic proofs to patients using writing hashes in the EOS blockchain (Grigg, 2017).

Coral health.

Coral Health, CHT token, is an Ethereum-based data sharing platform for EHR and genetic test results for personalized medicine (Abul-Husn & Kenny, 2019). With the usage of a precision medicine program, more successful results with lesser side effects could be obtained in treatment. Coral Health aims to create an interoperable, accessible, secure, and scalable healthcare ecosystem for fixing the issues in genetic data collection and data sharing (Park et al., 2017). In the system, patients can share their medical records directly with providers, laboratories, and others. During this process, they take control of their data access permissions using their accounts. Like Apple Health, the Coral Health system uses SMART and Fast Healthcare Interoperability Resource (FHIR: http://hl7.org/fhir/) protocols for establishing a connection between each mobile device of patients and other environments that host their medical data.

When a laboratory result or prescription is obtained for a patient, the notification comes to the patient’s mobile device. If a patient verifies the notification, pharma companies or others can buy the sample results. Information on patients is stored in an encrypted format on the mobile devices in a Health Information Trust Alliance (HITRUST) and the Health Insurance Portability and Accountability Act (HIPAA: https://www.hhs.gov/hipaa/index.html) compliant manner (Bosworth, Kabay & Whyne, 2014). Only the patient has a key for sharing the data. Besides, if one person arrives in an emergency room and if the patient is unconscious, the doctors can access the patient’s medical records fastly. Using the genetic information of this person and other biomarkers, the doctors can apply a personalized treatment. The system supports different data types such as images from radiology, genetic test results, or microbiology test results.

Patientory.

The Patientory, PTOY token, is an Etheruem-based electronic health record data-sharing platform. In the classical system, a patient can receive his/her medical history and can share it with doctors; or a doctor can follow the health status of the patient with a mobile healthcare application. However, in this classical system, those processes take time and doctors can only access limited data. Also, there is a centrality problem in classical systems, which means that classical systems are not completely secure. In the proposed system, users can access their medical records easily, and also they can update the data or share it with other scientists or doctors. When the system is developed, primary attention was paid to the Health Insurance Portability and Accountability Act law. The patient-related application is free, and 10 MB storage capacity is provided to the participants. After 10 MB of data storage, users are required to pay a certain amount of PTOY. Patients can create individual profiles via their mobile applications. They store the patients’ information on a secure, HIPAA-compliant blockchain platform. At the same time, they communicate with other users who have a similar health problem. As in all other projects, it uses a smart contract for data access permissions of users that can be easily edited. The system is international, so it is not limited to use in the USA (Mcfarlane et al., 2017).

Medicalchain.

Medicalchain, MTN token, is an Ethereum and Hyperledger based EHR sharing project that empowers secure, quick and transparent transactions and enables the utilization of medical information. It is built by employing a dual blockchain structure. The first blockchain controls the access to health records and it is built by using Hyperledger Fabric. The second blockchain, Ethereum, is fueled by underlies all the applications and services for the platform. The Hyperledger is a permission-based network, which means that someone needs to sign up to join this network. For specific applications which require keeping the data confidentially, or which only allow particular people to access the data, Hyperledger Fabric is used. Hyperledger Fabric accommodates numerous layers of permission, which means that there are several methods to adjust access control. Hence, Hyperledger Fabric could be a better solution for inspecting the access to health records.

One of the advantages of using a smart contract in the healthcare system can be exemplified as follows. Doctors spend so much time on billing and insurance-related actions. If these processes were achieved with smart contracts, and if these processes were validated by Ethereum, significant savings in terms of cost could be achieved. In order to prevent the attempts of stealing the identities, the system is partnered with Civic (CIVIC Technologies, 2017). The system uses Civic’s authentication services, where the Civic securely manages identities using a decentralized network. They use the biometrics of users for the verification of identities. Participants consist of practitioners, patients, and research institutions. Symmetric key encryption is used to supply privacy and to encrypt health records (Agrawal & Mishra, 2012; Kumar, Munjal & Sharma, 2011). The Medicalchain also has a backup access system for emergency conditions. It uses a bracelet that the patients wear. To unlock and reach the information, two doctors would need to scan the bracelet. In this way, the doctors can easily reach the patient’s medical records and hence, the best treatment can be provided (Medicalchain, 2018).

GemOS.

The GemOS, which does not have a token yet, is Ethereum and Hyperledger based EHR sharing platform. GemOS is developed to support patient-centric healthcare and personalized medicine with the partnership of Philips. Siloed databases mainly consist of electronic health records. Different companies or organizations manage each one and people has to spend money on reconciliation. The project aims to isolate sensitive data, create scalability, flexibility, and present an extensible platform to solve fundamental issues like reconciliation and others. It provides an explicit mechanism based on four essential components; data, identity, network, and logic. Although detailed technical explanations are not available for this project, to get an idea of this project’s overall structure, interested readers can refer to their white papers (Kannan & Smith, 2016).

e-Estonia.

The e-Estonia is a Keyless Signature Infrastructure (KSI: https://www.guardtime-federal.com/ksi/) blockchain-based project that adapts the registration of many state institutions like the electronic healthcare system, in cooperation with Guardtime. Figure S2 illustrates the KSI technology of Guardtime. Keyless Signature Infrastructure (KSI) has been developed in Estonia. While preserving data privacy, it is used internationally to ensure that networks, processes and data are free of compromise (KSI Blockchain Website). Unlike traditional digital signature approaches, KSI uses only hash-function cryptography. In KSI, verification only relies on the security of hash functions and the availability of a public ledger. This is commonly referred to as a blockchain. KSI blockchain claims that it can get over scalability and settlement time problems of traditional approaches. Conventional approaches to blockchain expand linearly with the number of transactions. On the other hand, the KSI blockchain grows linearly over time and it is irrespective of the number of transactions. The distributed consensus protocol of the KSI blockchain is limited. By limiting the number of participants, consensus can be achieved synchronously, removing the need for PoW and ensuring that settlement will occur within a second. KSI Blockchain is also using hash trees to keep the data. All transactions within a time frame are collectively saved as a hash tree to support the high number of signatures. The critical values at the top of the trees are connected to each other to create a general hash tree.

The system has two main applications. The first one is electronic patient records and the second one is electronic prescription. Electronic patient registration system combines different types of health information from the institution. The blockchain controls safety of patient records and access to the log of data. Also, patients can access their own records using the given authorization for access. The system supports the usage of mobile devices (e Estonia, 2012). All prescriptions have been completed in the digital environment. With e-prescription, only using the ID-card number of a patient is enough to reach the original prescription. Besides, it is not needed to continuously visit the doctor if a patient is using a particular medicine for a long time. Through the system, the patient notifies the doctor with a request; then, the doctor uploads a new e-prescription to the system (Buldas A. Kroonma A. Laanoja, 2013).

Blockchain development era in EHR sharing

In the period from 2019 to the present day, academic studies are carried out on this subject. It is seen that in the blockchain development era, cloud-based and encryption-based complex applications are mainly proposed. At this stage, (i) studies have focused on designing a blockchain-based healthcare system as in the first age. Although the detailed description of the system structure is less than the first age, it is more than the third age, and (ii) instead of focusing on any disease prediction and patient monitoring method, the studies generally focused on the questions of how to use different encryption techniques in systems, how to implement the blockchain-supported modules and evaluate performance.

Encryption & blockchain based medical data management

Liu et al. (2019) suggested a decentralized access control method among the healthcare providers by using proxy re-encryption wherein each healthcare provider stores the patients’ EHRs in a private and permissioned blockchain. In the proxy re-encryption method, one party A entrusts a trusted third party to transform the ciphertext encrypted with its public key into ciphertext encrypted with the other party B’s public key. Then, B could decrypt the ciphertext with its own private key, i.e., the data sharing is realized. Therefore, the encrypted data becomes very secure, and A’s private key does not have to be disclosed. The platform is composed of three parties as system manager, patient and hospital. The system manager (SM) is a trusted third party that the health department operates, and it is responsible for generating public/private keys for healthcare providers along with the re-encryption key based on the requester’s public key. When a hospital physician requests access to all available EHRs for a patient in existing healthcare provider blockchains, the SM retrieves the EHRs from local blockchains, produces the encryption key using the receiver’s public key, and re-encrypts the EHRs before returning them to the physician. The dPoS consensus protocol was strengthened by designating the SM as a specific component to evaluate published records by physicians to raise or decrease physicians’ credit scores to reduce the computation and communication costs of local blockchain systems. They also suggested a Symptoms-Matching method to enable patients who are enrolled in various healthcare systems but have the same disease symptoms to communicate with one another using a protected session key. PBC and OpenSSL libraries are used to execute the proposed scheme. Finally, the proposed scheme’s security and efficiency are assessed.

In the study of Niu et al. (2020), attribute-based encryption is used to meet the EHR managing system’s multi-user retrieval requirement and providing fine-grained access control for users. The basic concept behind this approach is to store encrypted EHRs in healthcare providers’ local cloud storage systems, while extract keywords are published on a permissioned blockchain. As a result, users can only reach the patient’s EHRs by obtaining their search trapdoors to run keyword searching through the blockchain. There are six entities as system manager, EHR system, patient, doctor, data user, and blockchain in the architecture. The system administrator is responsible for the entire system and generates public keys and private keys for each user. Also, the system administrator distributes and revokes the attribute as attribute authorization. EHR system refers to a user who provides medical services to a medical institution. Every hospital usually has a server and a number of computer clients (doctor). Each doctor keeps an encrypted copy of the patient’s medical details on the hospital’s servers. To achieve electronic health record sharing among hospitals, the server administrator stores some keywords of these records (in the form of transactions) in the permissioned blockchain. When a doctor uploads an electronic health record to the server, the server must confirm the doctor’s identity. For the performance evaluation, numerical experiment analysis of the proposed approach is made through a simulation. The implementation is based on Java pairing-based cryptography.

Cloud infrastructure & blockchain based medical data management

Due to the large size of electronic health records, keeping all of the data on the blockchain is inefficient. A cloud environment is basically a series of storage devices that are logically connected to form a vast infrastructure that provides services, including data storage and computing power. For this reason, such health data is usually stored in clouds with large storage capacity, while transaction records and important metadata are processed into the blockchain. As a result, the number of studies using cloud and blockchain technology as a hybrid increases (IBM’s Medical-Blockchain; Al Omar et al., 2019).

IBM (https://github.com/IBM/Medical-Blockchain/blob/master/README.md) proposes a blockchain-based medical data management platform. This platform is based on its own blockchain platform. In the architecture, data is not saved directly to the blockchain. Instead, while medical data are stored on off-chain, called the Redis Database, data management processes are saved on-chain. It has four types of roles. The first one is the solution admin, who is a manager of hospitals. It is responsible for recruiting and assigning a new hospital. The second one is hospital admin, which is responsible for adding and managing a new patient or doctor to the hospital as a user. Other users are patients and doctors, respectively. While doctors can perform operations such as seeing and uploading their patients’ documents, patients regulate their access permissions to this data. Since the project is a ready platform, it allows individuals to set up their own systems by following the detailed instructions given on the project site. It contains multiple IBM technologies and is a Hyperledger Fabric distribution.

Al Omar et al. (2019) proposed a blockchain-cloud-based EHR sharing platform called MediBChain to preserve privacy and ensure pseudonymity. According to architecture, the private healthcare data in the cloud is controlled by only the patient herself. The main idea of this work is to keep the sensitive healthcare data on the blockchain to attain accountability, integrity, and security. Present healthcare systems lack pseudonymity as those only stores the data in the cloud, but this platform ensures the pseudonymity of patients. Pseudonymity is achieved by using cryptographic functions. The platform consists of 5 essential system components: data sender, data receiver, registration unit, private accessible unit, and blockchain. Data sender is the patient, who will send her encrypted healthcare data to the system. Encryption of data will be done at the very beginning of MediBchain’s process execution. Data receiver is the doctor, who will request the data after authenticating itself to the system. Registration Unit is an authenticator who will store the ID and password of the Users to be used further. Finally, PAU is the intermediary unit that provides communication between the blockchain and its users. After authentication, system users interact with the PAU to send their data to the system using a secure channel. Blockchain holds the data of the users. Each transaction in the blockchain returns an identifier to help the users to access the data further. For the performance evaluation, the authors have implemented a smart contract using Solidity and shown different analogies of costs as transaction cost and execution cost. Then they have evaluated a Java implementation of input and output generation algorithm using Elliptic Curve Cryptography (ECC) for MediBChain.

Tanwar, Parekh & Evans (2020) addressed current healthcare market issues mainly data access management. To achieve safety and protection for patient data in the EHR framework, blockchain-based system architecture is proposed for access management policies. There are four participants in the proposed system as patient, clinician, lab, and system admin. Various assets and chaincodes, such as GrantAccess and RevokeAccess, are specified in this framework to grant or revoke permissions to or from a requester. Participants register through the client application or SDK, requesting an enrolment certificate via a Membership Service Provider (MSP). MSP is responsible for registering the patients and healthcare providers and generating the public and private keys.

Participants in the scheme have various responsibilities and can only view records to which they have been given access. Patients can add records using the client application, which invokes the chaincode for committing a transaction to the network. Providers, such as clinicians and laboratory staff, may use the network to query the data they need. If the patient permits the clinician or laboratory participant to display and update their documents through the EHR ledger network, the clinician or laboratory participant may view and update the patients’ permission records as requested. For better outcomes, the proposed system’s performance is evaluated using the Hyperledger caliper by configuring block size, block formation time, endorsement policy, and proposed optimization for measurement metrics such as latency, throughput, and network protection

Blockchain as a platform era in EHR sharing

In the period from 2019 to the present day, academic studies are carried out on this subject. In the blockchain as a platform era, blockchain is transformed into a platform and additional AI-based algorithms are running on the blockchain platform. This period can be considered as an initial steps of building a data ecosystem with the help of blockchain technology. During this period, (i) studies have focused on designing a blockchain-based healthcare system with some patient monitoring and disease prediction methods, and (ii) artificial intelligence methods have been integrated into the systems, focusing more on these issues and evaluating the performance of the systems from this perspective.

Artificial intelligence & blockchain based medical data management

The internet of things (IoT) is becoming more popular recently, with applications in a variety of fields, including healthcare. Due to the growing demands of the Internet of Things, a vast amount of sensing data is provided by various sensing devices. Artificial intelligence (AI) techniques are essential for real-time data analysis that is both scalable and precise. However, there are many obstacles to designing and developing a practical big data analysis methodology, including centralized infrastructure, protection, privacy, resource constraints, and insufficient training data. Blockchain technology, on the other hand, is gaining popularity because of its decentralized nature. It is promoted for removing centralized control and resolving AI problems by allowing secure sharing of data and resources among different nodes of the IoT network. For all these reasons, nowadays, it has been started to work on combining blockchain-based IoMT with AI techniques (Veeramakali et al., 2021; Połap, Srivastava & Yu, 2021; Chen et al., 2021).

In this study (Veeramakali et al., 2021), IoT systems are used to gather data from users at first. The OPSO algorithm is then used to share secret images. The HVE-NIS algorithm is then used to hash value encryption. Finally, using the ODNN-based diagnostic method, the disease is diagnosed. Image steganography conceals the hidden message in the cover is and sends the private message from sender to receiver to a larger group of people. The OPSO method is used to integrate a private image into cover images in the first step of this approach. The positions are precisely chosen during embedding in order to improve the PSNR of the cover images. The HVE-NIS model used in the second level is a character-encoding system that works by traversing data using 0s and 1s. It uses valid data from neighboring bits of the input character to assign the smallest code words for each character in the input sequence. OPSO-DNN-based medical diagnosis model has operated in the last step using the strategy of ANN along with various hidden and output units known as DNNs. At the time of implementation, a DNN is made up of pre-training and fine-tuning stages. In addition to the traditional method, the OPSO algorithm is used in this study to tune the parameters of the DNN and improve classification efficiency during the fine-tuning stage of the DNN. Performance evaluation was made with a simulation on these issues in order to emphasize the novelty of the study. During the diagnostic process, the OPSO-DNN model gave high results for the parameters of sensitivity, specificity and accuracy. Similarly, the HVE-NIS model achieved the best compression of blockchain hash values in terms of compressed file size, compression ratio and space-saving.

Complex tasks can be broken down into individual cooperating objects using multi-agent structures. This form of architecture is now being used more frequently to develop applications for the Internet of Medical Things (IoMT). In Połap et al.’s study (Połap, Srivastava & Yu, 2021), an agent architecture for IoMT is proposed. This architecture focuses on the use of machine learning with the policy of exchanging selected data or trained classifier models, as well as user data security on the blockchain. We can list the features of this new multi-agent system architecture as follows, respectively: (i) allows separating specific tasks to agents’ units, (ii) agents combine with federated learning, (iii) uses consortium mechanism for classification results from many machine learning solutions, and (iv) performs sharing and protecting private data processes with blockchain.

In this system, there are three types of agents and users. Agents are data management agents (DMA), indirect agents (IA) and learning agents (LA), respectively. The cloud infrastructure is used in this study because it allows various agents to access databases from different locations. In the architecture firstly, LA is proposed on some devices with a connection to databases with medical results considering data confidentiality. These learning agents combined with federated learning (Peng et al., 2021) due to the number of threads which allows for parallel training of classifiers and grouping their weights to obtain one classifier architecture. For training purposes, all results should be labeled first. Over time, the network will become more accurate and be able to take over the labeling task. Having trained models, the results can be shared on the database. Other agents can use this model for immediate data classification and analysis. IA will be responsible for making the full diagnosis of the individual patient. The results can be updated on the database for use by other agents (for instance, for training purposes) and send further. DMA manages the data as well as uses the other results. His communication is about supervising or using others and communicating with patients or the doctor. This is the only type of agent that has private access to the blockchain.

The system users are medics, technicians, and patients, respectively. Patients can view information about themselves, and technicians can access databases to add new records. The last group is doctors who can access the databases through the DMA and ask for making other simulations/classification. All data are placed on databases in the architecture, and some of them are encrypted in the blockchain. In the first part of the performance evaluation, LA and IA and their operation in different environments are analyzed to show the potential in using them. Then, t accuracy, processing time, and general performance are measured. After that, all agents are combined and simulated a small system to check the general performance and DMA operation. Finally, in this experiment, the delay of constantly training agents is analyzed and trained ones.

Chen et al. (2021) introduce a Blockchain-enabled diabetes disease detection system that uses various machine learning classification algorithms to provide earlier detection of the disease and secures the patients’ EHRs. The patient’s health information is collected through wearable sensor devices in this EHRs sharing network, which combines symptom-based disease prediction, blockchain, and the interplanetary file system (IPFS). The system consists of multiple structures. These are registration center (RC), EHR manager, IPFS, ML model and administration unit (AU). In the system, the patient and doctor first register by submitting a request to the Registration Center (RC), which collects all pertinent information and assigns a private key and ID, which is then forwarded to the Administration Unit.

The EHR manager acts as a central controller and performs a variety of tasks. When a user wants to conduct a Blockchain transaction, they should contact the EHRs Manager. Whenever a patient or doctor makes a request, the EHRs manager requests the requester’s public key. After supplying the public key, it is sent to the administration unit for verification, and it is determined if the requester has the right to upload or retrieve data to/from blockchain based on the public key. The administration unit uses a smart contract from the policy list to validate the requester’s public key. When a patient or practitioner is successfully validated, the EHRs Manager sends an encrypted transaction to the IPFS, a cloud storage server, in order to create a link with the Blockchain network, using the EHRs Manager’s public key. Finally, various performance measurement metrics such as accuracy, sensitivity, specificity, precision, f1-measure, Matthew correlation coefficient (MCC), and ROC curve are used to compare the performance and efficiency of the prediction models.

Multi-modal & blockchain based medical data management

Arul et al. (2021) focus on problems titled the confidentiality of patients’ health data transmission and privacy in IoMT devices. Accordingly, the Multi-Modal Secure Data Dissemination Framework (MMSDDF) for secure patient data access and control has been proposed in this paper, which is based on blockchain in IoMT. According to this architecture, there are mainly two types of actors as doctors and patients. In the first stage of the functioning mechanism, patient data from IoMT devices is collected. However, due to the fact that these data are real-time and their size is vast, it is not possible to keep all of them on the blockchain. For this reason, while patients keep their private data in the off-chain database, the data that needs to be processed in the blockchain is kept in the blockchain. Thanks to this procedure: (i) doctors can concurrently conduct data analysis and share the results with those with access to this information using the blockchain in IoMT, (ii) blockchain’s key has been used to a healthcare application network where the patient’s health data can create warnings that are significant to authenticated healthcare providers securely.

Three types of transactions have been successfully used in this model. The first transaction creates a session key, which enables a user to keep track of all access. The session key encrypts the data and transfers it to the cloud, while the public key encrypts the session key. After that, the network receives the encrypted session key but the session key can only be decrypted with the patient’s key. Thus, health data is only accessible to nodes with access to the patient’s data session key, which can be saved in smart contracts. The second transaction occurs if a patient requests that a doctor can view their data. The patient decodes the network’s session key with her or his key. The doctor will use the session keys to access the patient’s medical records using this method. The specialist will even have to supply the patient with a new medical record. It will use a newly created session key and encrypt the session key with the patient’s public key, so the patient will use his private key to access the session key. The accuracy ratio, prediction ratio, response time, delay time, and latency range of the proposed MMSDDF system have all been considered for the performance evaluation. The data processing has been analyzed for 2 s in this simulation setting, with various response ranges of time.

Discussions

The advancements in the next-generation sequencing technologies resulted in the generation of hundreds of gigabytes in a single run, and up to two billion human genomes are expected to be sequenced in the next ten years (Koboldt et al., 2013; Navarro et al., 2019). In addition to genomic data, high-throughput technologies generate different types of -omics data in high quantities, whose management, analysis, and storage processes require specific infrastructures and pipelines. While sharing these -omics data offers the unique opportunity to increase our knowledge by obtaining new information from the re-analysis of the same datasets and collective datasets, it imposes several challenges of ethical, legal, and technical nature. In this respect, recently, blockchain technology has picked up significant attention in diverse fields, including genomics, since it offers a new solution for these problems from a different perspective (Yli-Huumo et al., 2016; Casino, Dasaklis & Patsakis, 2019). To overcome the data sharing problems in genetics and EHR, several blockchain-based projects have been developed, yet there are only a limited number of studies that combine the cryptocurrency system and the academic studies. Thus, we believe that this review (i) could be very timely in the sense that it reviews several blockchain-based applications in genomics and healthcare both from an academic perspective and from an industrial perspective, and (ii) can show readers in this area how the use of the blockchain has changed over time. Our review article focuses on data sharing projects in 2016 and today that concentrate on genomics and electronic health records. While reviewing the projects, we divided the systems into two main topics: genomic data sharing and EHR sharing. While genomic data sharing studies are respectively Nebula Genomics, Zenome, Genecoin, Gene-Chain and DNATIX, the other studies; Medrec, IRYO, Coral Health, Patientory, Medicalchain, GemOS, e-Estonia, Liu et al. (2019), IBM’s Medical-Blockchain, Al Omar et al. (2019), Tanwar, Parekh & Evans (2020), Niu et al. (2020), Veeramakali et al. (2021), Połap, Srivastava & Yu (2021), Chen et al. (2021) and Arul et al. (2021) are EHR sharing studies. To demonstrate the evolution of blockchain in EHR sharing systems, we divided the timeline into three parts: (i) proof of concept era, (ii) blockchain development era, and (iii) blockchain as a platform era. It is seen that in the proof of concept era, there are several proof-of-concept applications that require core development. In the blockchain development era, we observe cloud-based and encryption-based complex applications are proposed. In this time block, the cloud infrastructure is improved and some additional analyzing tools are coupled with blockchain as building data commons. Blockchain is transformed into a platform and additional AI-based algorithms are running on blockchain platforms in the blockchain as a platform era. It reveals another potential of blockchain that can be helpful while designing the data ecosystem in the field. In Tables 2 and 3, we comprehensively evaluate these projects.

Table 4 specifies the common advantages and shortcomings of the current projects. The main advantages of using blockchain in these platforms are; (i) access permissions of data are controlled by data owners, (ii) analysis costs can be reduced, (iii) communications between the data owner and buyer accelerates and it becomes transparent, (iv) data collection process is accelerated and (v) privacy problems are partially solved. However, there are still unresolved problems such as (i) full anonymity cannot be provided, (ii) there is no preventive system for attack scenarios, (iii) key related problems, (iv) energy consumption and scalability problems of blockchain technology and (v) no detailed documentation. We point out the importance of data sharing and analysis for genomics and EHR, and we reveal how blockchain technology fixes the potential difficulties in this field. In addition to our brief introduction to projects, we present a novel classification scheme by identifying the common metrics and technical differences in Tables 5 and 6, Figs. 4 and 5. When examined in Table 2, the main features of projects in the proof of concept era are seen. Among these 12 projects, only five of them are genomic data-sharing platforms and after 2018, all work has been completely focused on EHR sharing. While Nebula Genomics, Zenome, Genecoin, DNATIX, Medrec, Coral Health, and Patientory are based on Ethereum, Gene-Chain is based on Hyperledger, IRYO is based on EOS, e-Estonia is based on KSI, and Medicalchain and GemOS are based on a combination of Ethereum and Hyperledger. All of the proposed projects run on smart contracts. Hence, it can be concluded that Ethereum or Hyperledger technologies are preferred by projects which are developed between 2016–2018. Generally, Ethereum is used by genomics-related projects, contrary, healthcare-related projects use Hyperledger.

Table 4 The common features of projects.

Advantages	• Data owners control data access permissions
•  Easily and directly communication
• Metadata are stored on a block instead of original data
• Quick data transmission
• Data standardization
• The immutable and distributed ledger of transactions
• No intermediary companies
• Reducing analysis costs
• Verification mechanism
• Providing interoperability
• Pseudoanonymity	
Disadvantages	• No fully homomorphic encryption, so shared data is not in a fully encrypted format
• There is no utterly preventive system towards attacks
• No fully anonymity; only pseudo-anonymity
• No exact scalability solutions
• Key challenges
• Energy consumption	

Table 5 The unique features of projects in proof of concept era.

The unique feature of projects	Advantages	Shortcomings	
Nebula Genomics
(Grishin et al., 2018)	Nebula sequencing facility, Using homomorphic encryption. Supporting of third-party apps. Data owner earns profit	No supporting more unstructured phenotyping data and medical data, system for only a human organism	
Zenome
(Kulemin, Popov & Gorbachev, 2017)	Supporting non-human organism data. Data owner earn a profit. System use notifications for users	Genomic data is shared without encrypted format	
Genecoin
(Schorchit et al., 2018)	Claiming an equal and fair system	Genomic data is shared without encrypted format	
Gene-Chain
(Encrypgen, 2017)	The system has strict verification procedures.	Genomic data is shared without encrypted format	
DNATIX
(DNATIX, 2018)	The system has its own data compression algorithm and virtual machine	Data length limitations and there is no clear information about providing	
Medrec
(Azaria et al., 2016)	System use notifications, Supporting patient monitoring, metadata are stored on the block. Private blockchain	Should be ensured that the name of the patient is completely confidential	
IRYO
(IRYO, 2017)	Data owner earn a profit, Supporting, disease prediction and patient monitoring, System can study with anonymous data and use notifications	The analysis cost does not decrease much	
Coral Health
(Park et al., 2017)	Interoperability. System use notifications. Supporting both EHR and genetic test results	The cost for storage is high, and sensitive data is shared without encrypted format	
Patientory
(Mcfarlane et al., 2017)	Suitable for HIPAA -compliant, The system is international not special only the USA	After 10 MB storage area, another fee is charged	
Medicalchain
(Medicalchain, 2018)	The system has numerous layer of permission for verification identities of users. For the emergency condition it has an EHR-supported bracelet device. Supporting biometric data	The possibility of fake biometric data’s use and the bracelet being stolen by anyone	
GemOS
(Kannan & Smith, 2016)	The extensible platform, Solving the reconciliation issues	There is no clear information about data privacy	
e-Estonia
(e Estonia, 2012)	The system punishes people who upload wrong data. In an emergency case, doctors can read patient’s EHR using its ID. System use notification. Instead of RSA, post-quantum signatures are used	Robustness of ID’s security	

Table 6 The unique features of projects in blockchain development era and blockchain as a platform era.

The unique feature of projects	Advantages	Shortcomings	
Liu et al. (2019)	With proxy re-encryption, proposing a decentralized data management across hospitals when EHRs are recorded locally in hospital blockchains.	When storing EHRs on the blockchain system, there is a significant storage overhead.	
IBM’s Medical-Blockchain	Store private healthcare data off-chain and manage medical data using blockchain. Uses an unique approach for consensus.	It has got a complex architecture. Therefore, it should be considered in detail when integrating services.	
Al Omar et al. (2019)	User-centric EHR systems giving total control of data to users. Permissioned Blockchain and other functions restrict intruders from a security breach.	When storing EHRs on the blockchain system, there is a significant storage overhead.	
Tanwar, Parekh & Evans (2020)	Creating a Hyperledger-based access control system to handle EHRs safely.	Causes high storage overhead; Dependents on MSP.	
Niu et al. (2020)	Using an ABE to address the multi-user retrieval requirement of an EHR management system.	The system’s efficiency was only assessed based on the number of attributes and search time.	
Veeramakali et al. (2021)	Proposing an optimal deep-learning-based secure blockchain healthcare diagnosis model. It examines the model in three different stages in detail.	When storing EHRs on the blockchain system, there is a significant storage overhead.The blockchain side of the work is less processed than the other parts.	
Połap, Srivastava & Yu (2021)	Allows separating specific tasks to agents units. Agents combine with federated learning. Performs sharing private data processes with blockchain.	The blockchain side of the work is less processed than the other parts. There is no analysis of the offer in terms of security.	
Chen et al. (2021)	Blockchain based system uses various machine learning classification algorithms to provide an earlier detection of the disease and secures the patients’ EHRs.	There is no performance evaluation of the blockchain for the platform.	
Arul et al. (2021)	Proposing multi-modal healthcare data dissemination framework using blockchain in IoMT. Data optimization and management were examined in detail.	Detailed information about how users got into the system was not given.	

As shown in Fig. 4, Nebula Genomics, Genecoin and Gene-Chain support register kit. It means that they have their own sequencing facilities. In Nebula Genomics, IRYO and Zenome, data owners earn money. While Zenome supports both human and non-human data, other projects only support human data. Only Nebula Genomics share data in encrypted format using partially homomorphic encryption. It means that data privacy is better than others in Nebula Genomics. IRYO and Zenome are included in disease prediction metrics. Because they apply some artificial intelligence methods on data and obtain predictions related to diseases. While Medrec, IRYO, Coral Health, Patientory, Genecoin, Medicalchain, GemOS and e-Estonia have mobile applications; Medrec, IRYO, Coral Health, Patientory, Medicalchain and e-Estonia have a patient monitoring system. Finally, we need to mention that the studies between these dates are project-based and the academic studies have been proposed after this stage.

When examined in Table 3, the main features of projects in the blockchain development era and blockchain as a platform era are seen. Among these 9 studies, while Liu et al. (2019), IBM’s Medical-Blockchain, Al Omar et al. (2019), Tanwar, Parekh & Evans (2020) and Niu et al. (2020) are in the blockchain development era; Veeramakali et al. (2021), Połap, Srivastava & Yu (2021), Chen et al. (2021) and Arul et al. (2021) are in the blockchain as a platform era. Considering all the work, they are designed as private blockchains. The blockchain platform and consensus algorithms used are not specified in most studies. However, careful processing of these metrics in a blockchain-based system provides more detailed information about the system. As shown in Fig. 5, while Liu et al. (2019), Al Omar et al. (2019), Tanwar, Parekh & Evans (2020) and Veeramakali et al. (2021) store data on-chain, others do not (off-chain). When storing EHRs on the blockchain system, there is a significant storage overhead. If we compare the proof of concept era with the blockchain development and blockchain as a platform era, we obtain the following conclusions: (i) while the proof of concept era covered both genomic data sharing and EHR sharing, in other ages, studies are done entirely on EHR sharing, and (ii) while in the proof of concept era, each project designed the detailed architecture of blockchain-based healthcare systems, in other ages, the blockchain details are reduced and the proposed methods mostly works with some other techniques. If we compare the blockchain development era with the blockchain as a platform era, we obtain the following conclusions: (i) in the blockchain development era, studies have focused on designing a blockchain-based healthcare system as in the first age. However, the main focus is not always the detailed description of the system, and (ii) at the blockchain development era, instead of focusing on any disease prediction and patient monitoring methods, the studies generally focused on to improve the infrastructure with additional cloud management mechanisms or encryption techniques, (iii) in the blockchain as a platform era, studies have focused on designing a blockchain-based healthcare system with some patient monitoring and disease prediction methods, and (iv) at the blockchain as a platform era, different artificial intelligence methods have been integrated into the systems, focusing more on these issues and evaluating the performance of the systems from this perspective. Finally, from the beginning to the present, we see that the blockchain will be used as a platform in the future.

Figure 4 Classification of projects in proof of concept era.

Figure 5 Classification of projects in blockchain development era and blockchain as a platform era.

Conclusions

Blockchain-based platforms have a snowball effect with their potential to solve several security and agreement issues, including data sharing and secure computing in a public network. In addition to the published scientific papers, several applications of blockchain technology are already implemented and their cryptocurrencies are already available on the market. Therefore, it is not easy (i) to understand the trends for cryptocurrencies and the usage of blockchain in a specific field, and (ii) to filter/sort the existing applications with a technical background. Moreover, compared to the other studies, there is no study showing the evolution of the use of blockchain in healthcare. In this respect, here we aim to review the blockchain-based applications in genomics and healthcare both from an academic perspective and from an industrial perspective. Unlike existing studies, we wanted to cover cost analysis, ownership, data collection, authorization, security, and anonymity issues. In this field, there is no such comprehensive review available to the best of our knowledge, and we intend to fill in this gap via analyzing the existing blockchain-based EHR and genomic data sharing projects in detail and classifying them using different metrics. In addition to our brief introduction for these projects, we presented a novel classification scheme by identifying common metrics and technical differences. We have shown the advantages/disadvantages of these selected projects and discuss their key features.

In summary, to facilitate the diagnosis, monitoring and therapy of diseases with the effective analysis of -omics data in addition to other available data types, through this review, we put forward the possible implications of blockchain technology to life sciences and healthcare. We think blockchain could be used efficiently for operating regulations of Decentralized Autonomous Organizations such as The Cancer Genome Atlas (TCGA), International Cancer Genome Consortium (ICGC), etc. Considering the changes in studies over time, we see that blockchain technology is now used as a platform in the health field and it can have a great impact on building a data ecosystem in the healthcare management systems. At this stage, although it seems efficient to benefit from the advantages of blockchain technology in security and privacy issues, it should be considered in terms of energy and computation that the technology provides a much less advantage compared to the benefits of other methods used in the system. In this respect, very few studies evaluate how much blockchain technology needs in health systems in terms of performance. Other existing studies make comparisons of other methods used. In future studies, while blockchain technology is used as a platform with other methods (AI, cloud computing and edge computing), performance evaluation of the work should be done with different and meaningful metrics, and the necessity of blockchain technology in the system should be supported with these results.

Supplemental Information

Supplemental Information 1 The architecture of the IRYO system

Click here for additional data file.

Supplemental Information 2 KSI technology of Guardtime

Click here for additional data file.

Additional Information and Declarations

Competing Interests

Author Contributions

Data Availability

Burcu Bakir-Gungor is an Academic Editor for PeerJ.

Beyhan Adanur Dedeturk conceived and designed the experiments, performed the experiments, analyzed the data, prepared figures and/or tables, authored or reviewed drafts of the paper, and approved the final draft.

Ahmet Soran conceived and designed the experiments, analyzed the data, authored or reviewed drafts of the paper, and approved the final draft.

Burcu Bakir-Gungor conceived and designed the experiments, authored or reviewed drafts of the paper, and approved the final draft.

The following information was supplied regarding data availability:

This is a review article; there are no raw data or code files to share.

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
