# Peer review of "Blockchain for genomics and healthcare: a literature review, current status, classification and open issues"

_PeerJ, doi:10.7717/peerj.12130_

## Round 0.1 · original submission · Major Revisions

There are several issues with this paper. Firstly, this review misses many relevant papers. It has been pointed out by three reviewers. Reviewer 2 believes that most of the papers are outdated and the authors should analyze mainly the solution in the last 2 years (2021 and 2020). Reviewer 4 also suggests adding more papers especially those in 2021 and 2020. Secondly, there is a lot of room for improvement in the presentation, including the organization of the paper (see comments of reviewers 1 and 2). In addition, some in-depth analysis needs to be added. Please consider the suggestions made by reviewer 4.

Reviewer 1 ·

Basic reporting

No comment

Experimental design

No comment

Validity of the findings

No comment

Additional comments

In this paper, the authors aim to highlight the challenges of EHR and genomic data sharing. They also attempt to answer "Why" or "Why not" the blockchain technology is suitable for genomics and healthcare applications in detail.

I think overall, the readability of the paper should be improved. There are many issues in verb tense and noun usage with overuse of "the" common from non-native speakers. I would recommend that the paper be professionally proofread.

Content wise, the following revisions are needed:

Subsections should also be numbered

Some sort of publication analysis should be presented early on in the survey with some insight into publications.

In general, more of the info presented should also be summarized in a collection of tables, flow charts or images. These items should be placed inline for reviewers to see in the revised version.

The paper lacks any proper organization in sections/subsections etc this should be addressed

The images given at the end of the paper are all of very low quality, please improve both quality of the image plus the actual presentation of them.

The tables (first column) should also give reference as to where to find more info about the item listed

The references and literature search has missed many papers in EHR and blockchain, see here some examples

Ramachandran S, et al. A Review on Blockchain-Based Strategies for Management of Electronic Health Records (EHRs). In2020 International Conference on Smart Electronics and Communication (ICOSEC) 2020 Sep 10 (pp. 341-346). IEEE.

Połap D, et al. Agent architecture of an intelligent medical system based on federated learning and blockchain technology. Journal of Information Security and Applications. 2021 May 1;58:102748.

Srivastava G, et al. The future of blockchain technology in healthcare internet of things security. Blockchain Cybersecurity, Trust and Privacy. 2020:161-84.

Reviewer 2 ·

Basic reporting

In the paper, the authors analyzed some basic solutions in the blockchain. I cannot agree that this paper should be accepted in the current form. The authors focused mainly on older solutions and as a survey, they should analyze mainly the last 2 years (2021 and 2020). Therefore in my opinion this paper is outdated.

Experimental design

Rewrite the paper to the current state of research, focus also on the application in practical solutions with machine learning like federated learning, etc. It is the latest approach in medical blockchains.

Focus also on practical differences between solutions and obtained results. Add some tables/figures to show more details in the literature. Compare the results, make some statistics, etc.

Validity of the findings

Add also more complex detailed discussion about future recommendations and compare your information with a paper:
Blockchain for healthcare data management: opportunities, challenges, and future recommendations, Neural Computing and Applications.
Write what exactly is the difference between your paper to this one.

Additional comments

In the paper, the authors analyzed some basic solutions in the blockchain. I cannot agree that this paper should be accepted in the current form. The authors focused mainly on older solutions and as a survey, they should analyze mainly the last 2 years (2021 and 2020). Therefore in my opinion this paper is outdated.

Rewrite the paper to the current state of research, focus also on the application in practical solutions with machine learning like federated learning, etc. It is the latest approach in medical blockchains.

Focus also on practical differences between solutions and obtained results. Add some tables/figures to show more details in the literature. Compare the results, make some statistics, etc.

Add also more complex detailed discussion about future recommendations and compare your information with a paper:
Blockchain for healthcare data management: opportunities, challenges, and future recommendations, Neural Computing and Applications.
Write what exactly is the difference between your paper to this one.


See these papers from this year:
1) VFChain: Enabling Verifiable and Auditable Federated Learning via Blockchain Systems, IEE Transactions
2) Agent architecture of an intelligent medical system based on federated learning and blockchain technology, Journal of Information Security and Applications
3)An intelligent internet of things-based secure healthcare framework using blockchain technology with an optimal deep learning model, Journal of Supercomputing
4)Multi-modal secure healthcare data dissemination framework using blockchain in IoMT, Personal and Ubiquitous Computing

·

Basic reporting

BASIC REPORTING
The title, abstract, introduction, methods, results and discussion are appropriate for the content of the text. Furthermore, the article is well constructed, the experiments are well conducted, and analysis is well performed. The figures are relevant, high quality, well labelled and described.

Experimental design

EXPERIMENTAL DESIGN
This is a review paper, so no experimental design. However, the discussion is original and the research is within the scope of the journal. Research question is well defined, relevant and meaningful. The overview and their proposal for a more suitable technology is highly technical, ethical and logistical.

Validity of the findings

VALIDITY OF THE FINDINGS
The introduction is comprehensive. The findings are meaningful. The conclusions are well stated and relevant to the research questions.

Additional comments

This paper was aiming at the challenges of EHR and genomic data sharing. The authors discussed the advantages and disadvantages of the blockchain technology implementation in genomics and healthcare applications. Furthermore, they proposed the general blockchain structure based on Ethereum, which is a more suitable technology for the genomic data sharing platforms.

Editorial Criteria
BASIC REPORTING
The title, abstract, introduction, methods, results and discussion are appropriate for the content of the text. Furthermore, the article is well constructed, the experiments are well conducted, and analysis is well performed. The figures are relevant, high quality, well labelled and described.
EXPERIMENTAL DESIGN
This is a review paper, so no experimental design. However, the discussion is original and the research is within the scope of the journal. Research question is well defined, relevant and meaningful. The overview and their proposal for a more suitable technology is highly technical, ethical and logistical.
VALIDITY OF THE FINDINGS
The introduction is comprehensive. The findings are meaningful. The conclusions are well stated and relevant to the research questions.

Overall, I think this review paper is novel and will be of interest to others in the community of omics data and EHR data sharing. This review paper does an excellent job outlining the urgent need to better manage the genomics data and further discussed the pros and cons of blockchain technology utilization in the field. In general, the work is convincing except some major and minor comments below:


Major Comments:

I’m wondering if it is worth mentioning that one of the key disadvantages of blockchain technology is the inefficiency of storing and querying data. And the computational efficiency is also low compared to traditional centralized databases.

I’m wondering if blockchain-based technologies support cloud computing and commonly used software services, tools & apps?

The words “data commons”, “data ecosystems”, “data cloud architecture” are really popular in the field of genomics data sharing. And they sound similar and are really confusing to researchers. Do you think it is worth adding those terms and explain a little bit?




Minor Comments:
The Figure 5 is hollow, and it is a little bit hard for the readers to differentiate the colors. I would recommend making it solid, with colors filled in.

Reviewer 4 ·

Basic reporting

- Paper is informal in many paragraphs; the authors must have corrected it.
- English must have improvements
- Recent resources must add to references, for example, 2020 and 2021 papers.
- The introduction section is too short to detail the motivations and challenges of this paper.
- Environmental features should be considered in terms of classification

Experimental design

-It should be compared with previous review papers. And its superiority over previous works should be determined.
-Healthcare environment characteristics must have more detail in this paper; it is necessary to identify the main features of the health care areas, a taxonomy or a framework must have provided, and based on that, the effects of using the block chain to become apparent. It can help in choosing methods in other articles or upcoming developments
- It is suggested that non-functional requirements are categorized, and the effects of using the block chain be illustrated.
-It is suggested that new articles that have been reviewed in recent years be reviewed to determine the advantages of the article over them.

Validity of the findings

- The simulation and implementation part of the article is examined and added to the tables, and converted into graphs if necessary
- The authors must Identify important parameters for evaluating each method or algorithm
- Examples given in the text should be transferred to health-related areas

Additional comments

This article should have a special advantage over other articles or previous reports that can be used later. For this reason, it is suggested that in some cases, the studies be done in more depth and detail and the author's analysis and conclusion be added to it.

---

## Round 0.2 · accepted · Accept

As the reviewers acknowledged, the revised manuscript addressed most concerns in the previous version. It is my pleasure to accept the current version for publication.

Reviewer 1 ·

Basic reporting

no comment

Experimental design

no comment

Validity of the findings

no comment

Additional comments

The authors have addressed and/or rebutted any comments or revisions I had requested in the previous round for this paper.

Authors should proofread paper carefully prior to final publication

Reviewer 2 ·

Basic reporting

The paper was inproved and it is rady for publication.

Experimental design

Ok

Validity of the findings

Expwrimental section was performed in proporcje way and shows great results.

·

Basic reporting

The title, abstract, introduction, methods, results and discussion are appropriate for the content of the text. Furthermore, the article is well constructed, the experiments are well conducted, and analysis is well performed. The figures are relevant, high quality, well labelled and described.

Experimental design

This is a review paper, so no experimental design. However, the discussion is original and the research is within the scope of the journal. Research question is well defined, relevant and meaningful. The overview and their proposal for a more suitable technology is highly technical, ethical and logistical.

Validity of the findings

The introduction is comprehensive. The findings are meaningful. The conclusions are well stated and relevant to the research questions.

Additional comments

This paper was aiming at the challenges of EHR and genomic data sharing. The authors discussed the advantages and disadvantages of the blockchain technology implementation in genomics and healthcare applications. Furthermore, they proposed the general blockchain structure based on Ethereum, which is a more suitable technology for the genomic data sharing platforms.

Major Comments:

I’m wondering if it is worth mentioning that one of the key disadvantages of blockchain technology is the inefficiency of storing and querying data. And the computational efficiency is also low compared to traditional centralized databases.

Response: Thank you for raising this important point. Blockchain can be very beneficial, but it does not mean that it is going to be a complete revolution. There are still too many pitfalls and improper parts of the blockchain, especially in healthcare management. We completely agree with the intent of this comment. Hence, this important point is mentioned in the revised manuscript. Please see Conclusion chapter on Page 21 of the revised manuscript. The disadvantages of each project are also discussed in the revised manuscript.

Feedback: Thanks for adding it to the revised version. The conclusion session and discussion session looks good to me.

I’m wondering if blockchain-based technologies support cloud computing and commonly used software services, tools & apps?

Response: Thank you for highlighting the relationship between cloud computing and blockchain. It is very common to use them to support each other. Blockchain is a distributed and decentralized system that works on a P2P network. Also, the smart contract mechanism is using its own virtual machine. When we build a proper design, it is possible to use software tools and/or apps on a blockchain platform. We can consider a blockchain application as a distributed app (Dapp). Recently, interoperability of different blockchain networks is also possible. Thus, it can be mentioned that blockchain-based platforms can support cloud computing or software tools distributedly. Please see the following answer as well.

Feedback: Thanks for the clarification. It totally makes sense to me.

The words “data commons”, “data ecosystems”, “data cloud architecture” are really popular in the field of genomics data sharing. And they sound similar and are really confusing to researchers. Do you think it is worth adding those terms and explain a little bit?

Response: Thank you for highlighting this important point. We believe that discussing these terms makes the manuscript more strengthen. We mentioned them in the Introduction section of the revised manuscript, please see Page 2. We also changed the organization in the revised manuscript and mention the evolution of blockchain technology in the healthcare management field. We discuss data commons and data ecosystems with blockchain timeline too. We believe that blockchain has a potential for contributing to data ecosystems.

Feedback: I strongly agree with you on this. And thanks for supplementing the introduction section. The new structure of the revised manuscript looks clear and straightforward to me.




Minor Comments:
The Figure 5 is hollow, and it is a little bit hard for the readers to differentiate the colors. I would recommend making it solid, with colors filled in.
Response: Authors fully agree with the intent of this comment and all figures are edited in the revised manuscript. Also, we would like to mention that some of the figures are totally changed in the revised manuscript, to address some other comments, e.g., removing stopped projects from the paper.

Feedback: Thanks for the updates for the figures. I also did see the comments raised by other reviewers in terms of the figures. The new set of figures look much more clear and informative. I don’t have any concern for the figures. Thanks!

Reviewer 4 ·

Basic reporting

No Comment

Experimental design

No Comment

Validity of the findings

1. The author should emphasize the advantages of this paper, more specifically, topic that introduced in this paper while not mentioned in other survey should be illustrated.
2. Open Issues must be classified and mentioned in the conclusion.

---

## Author Rebuttal · Round 0.2

# Responses to Reviewers for 57732

## Blockchain for genomics and healthcare: a literature review, current status, classification and open issues

### I. Rebuttal Letter Style:

We, the authors of 57732, first thank all the reviewers for their extremely thoughtful suggestions and their valuable time in reviewing our paper. We are very excited to have been given the opportunity to revise our manuscript. This revised version lists all detailed comments from each reviewer in bold and italic, followed by our corresponding responses and revisions. We have also given our revised manuscript as a marked-up copy (changes are highlighted in yellow); in addition to the final version of our revised manuscript. We also would like to note that the page numbers that we refer to in this rebuttal letter refer to the page numbers in the marked-up copy of our revised manuscript.

### Reviewer 1

Thanks for all the suggestions. In the revised manuscript, all comments have been carefully addressed and have been answered below respectively.

**(1) I think overall, the readability of the paper should be improved. There are many issues in verb tense and noun usage with overuse of "the" common from non-native speakers. I would recommend that the paper be professionally proofread.**

We have proofread the paper and edited the text accordingly. We improved the English to ensure that an international audience can follow the text.

**(2) Subsections should also be numbered.**

We thank the reviewer for this suggestion. In the revised manuscript, all sections and subsections are appropriately numbered.

**(3) Some sort of publication analysis should be presented early on in the survey with some insight into publications.**

We thank the reviewer for raising this critical point. According to this comment, the revised manuscript is updated in Section 3, page 6, 1st and 2nd paragraphs. In addition, in the revised manuscript, Figure 3 is also added to show the summary of publication analysis. Also, provided tables are presented to the reader, examining the studies in detail.

**(4) In general, more of the info presented should also be summarized in a collection of tables, flow charts or images. These items should be placed inline for reviewers to see in the revised version.**

As suggested by the reviewer, in the revised manuscript, in addition to the previous figures and tables, we provide additional materials. In the revised manuscript, Figures 3, 4, 5, and Tables 2, 3, 4, 5 and 6 contain summary information of the studies.

**(5) The paper lacks any proper organization in sections/subsections etc this should be addressed**

Thanks for this constructive feedback. We acknowledge and appreciate this comment. The sections/subsections are reorganized in the revised manuscript according to address this comment and some other comments from other reviewers.

**(6) The images given at the end of the paper are all of very low quality, please improve both quality of the image plus the actual presentation of them.**

Authors fully agree with the intent of this comment. In the revised manuscript, all figures are edited. Also, we would like to note that some of the figures are totally changed in the revised manuscript, to address some other comments, e.g., removing stopped projects from the paper.

**(7) The tables (first column) should also give reference as to where to find more info about the item listed.**

This is clearly an oversight on our part, we acknowledge and appreciate much. It is fixed in the revised manuscript; the references are given in the first columns.

**(8) The references and literature search has missed many papers in EHR and blockchain, see here some examples:**

- **Ramachandran S, et al. A Review on Blockchain-Based Strategies for Management of Electronic Health Records (EHRs). In2020 International Conference on Smart Electronics and Communication (ICOSEC) 2020 Sep 10 (pp. 341-346). IEEE.**
- **Polap D, et al. Agent architecture of an intelligent medical system based on federated learning and blockchain technology. Journal of Information Security and Applications. 2021 May 1;58:102748.**
- **Srivastava G, et al. The future of blockchain technology in healthcare internet of things security. Blockchain Cybersecurity, Trust and Privacy. 2020:161-84.**

Thank you very much for raising such an important point. In the revised manuscript, 9 new studies from 2019-2021 have been added. They are available in section 3.2 "Blockchain Development Era in EHR Sharing" Pages 12-15 and section 3.3 "Blockchain As a Platform Era in EHR Sharing" Page 15-18. In addition, the suggested references (Ramachandran et al 2020, Polap et al. 2021 and Srivastava et al. 2020) are also examined in the revised manuscript. After analyzing these recent studies, we changed the overall structure in the revised manuscript. Please note that the timeline has been created and the studies are discussed based on the blockchain evaluation timeline. We believe that this change makes the manuscript much stronger in the revision. We thank the reviewer for this suggestion.

## Reviewer 2

Thanks for all the suggestions. In the revised manuscript, all comments have been carefully addressed and have been answered below respectively.

**(1) The authors focused mainly on older solutions and as a survey, they should analyze mainly the last 2 years (2021 and 2020). Therefore in my opinion this paper is outdated.**

Thank you very much for raising such an important point. In the revised manuscript, 9 new studies from 2019-2021 have been added. They are available in section 3.2 "Blockchain Development Era in EHR Sharing" Page 12-15 and section 3.3 "Blockchain As a Platform Era in EHR Sharing" Page 15-18. The suggested references (Ramachandran et al 2020, Polap et al. 2021 and Srivastava et al. 2020) are also examined in the revised manuscript. After analyzing

these recent studies, we changed the overall structure in the revised manuscript. Please note that the timeline has been created and the studies are discussed based on the blockchain evaluation timeline. We believe that this change makes the manuscript much stronger in the revision. We thank the reviewer for this suggestion.

**(2) Rewrite the paper to the current state of research, focus also on the application in practical solutions with machine learning like federated learning, etc. It is the latest approach in medical blockchains.**

The revised paper is modified according to this comment. We thank the reviewer for this suggestion that makes the article more impactful. The latest approaches in medical blockchains are examined in the revised manuscript and 9 new studies from 2019-2021 have been added. They are available in section 3.2 "Blockchain Development Era in EHR Sharing" Page 12-15 and section 3.3 "Blockchain As a Platform Era in EHR Sharing" Page 15-18. Section 3.3 "Blockchain As a Platform Era in EHR Sharing" Page 15-18, focused on studies that use AI methods in conjunction with medical blockchains, and details are given.

**(3) Focus also on practical differences between solutions and obtained results. Add some tables/figures to show more details in the literature. Compare the results, make some statistics, etc.**

Authors fully agree with the intent of this comment. All studies have been carefully analyzed, but it is not possible to compare all systems in practice and theory. The reason is that the studies that emerged in the first years are only theoretical, and only a few of the latest studies have implemented blockchain-sided implementations. The studies conducted in recent years have used other methods, such as AI with blockchain, and evaluated system performance by these methods. We try to find the most suitable metrics that can be used for the comparisons. There is still not enough technical information presented in the articles to have statistics for the comparisons. We, finally, decide to show the blockchain evolution timeline to compare the similar type of projects within the group.

**(4) Add also more complex detailed discussion about future recommendations and compare your information with a paper. Write what exactly is the difference between your paper to this one:**

- **Blockchain for healthcare data management: opportunities, challenges, and future recommendations, Neural Computing and Applications.**

We thank the reviewer for this suggestion. In section 5, "Conclusions" Section (Page 21, 2nd Paragraph) of the revised manuscript, detailed discussion about future recommendations are added. In section 1.5 "Rationale of the Review and Intended Audience" Page 5, 1st Paragraph of the revised manuscript, we compared our study with many existing review articles, including the reference (Yaqoob et al. 2021). Also, we explained our differences from other articles in order to address this comment.

**(5) See these papers from this year:**

- **VFChain: Enabling Verifiable and Auditable Federated Learning via Blockchain Systems, IEE Transactions**
- **Agent architecture of an intelligent medical system based on federated learning and blockchain technology, Journal of Information Security and Applications**
- **An intelligent internet of things-based secure healthcare framework using blockchain technology with an optimal deep learning model, Journal of Supercomputing**
- **Multi-modal secure healthcare data dissemination framework using blockchain in IoMT, Personal and Ubiquitous Computing**

Thank you very much for these suggestions. Please see the answer given to the first comment.

In the revised manuscript, 9 new studies from 2019-2021 have been added. They are available in section 3.2 "Blockchain Development Era in EHR Sharing" Page 12-15 and section 3.3 "Blockchain As a Platform Era in EHR Sharing" Page 15-18. The suggested references (Ramachandran et al 2020, Polap et al. 2021 and Srivastava et al. 2020) are also examined in the revised manuscript. After analyzing these recent studies, we changed the overall structure in the revised manuscript. Please note that the timeline has been created and the studies are discussed based on the blockchain evaluation timeline. We believe that this change makes the manuscript much stronger in the revision. We thank the reviewer for this suggestion.

## Reviewer 3

Thanks for all the suggestions. In the revised manuscript, all comments have been carefully addressed and have been answered below respectively.

**(1) I'm wondering if it is worth mentioning that one of the key disadvantages of blockchain technology is the inefficiency of storing and querying data. And the computational efficiency is also low compared to traditional centralized databases.**

Thank you for raising this important point. Blockchain can be very beneficial, but it does not mean that it is going to be a complete revolution. There are still too many pitfalls and improper parts of the blockchain, especially in healthcare management. We completely agree with the intent of this comment. Hence, this important point is mentioned in the revised manuscript. Please see Conclusion chapter on Page 21 of the revised manuscript. The disadvantages of each project are also discussed in the revised manuscript.

**(2) I'm wondering if blockchain-based technologies support cloud computing and commonly used software services, tools & apps?**

Thank you for highlighting the relationship between cloud computing and blockchain. It is very common to use them to support each other. Blockchain is a distributed and decentralized system that works on a P2P network. Also, the smart contract mechanism is using its own virtual machine. When we build a proper design, it is possible to use software tools and/or apps on a blockchain platform. We can consider a blockchain application as a distributed app (Dapp). Recently, interoperability of different blockchain networks is also possible. Thus, it can be mentioned that blockchain-based platforms can support cloud computing or software tools distributedly. Please see the following answer as well.

**(3) The words "data commons", "data ecosystems", "data cloud architecture" are really popular in the field of genomics data sharing. And they sound similar and are really confusing to researchers. Do you think it is worth adding those terms and explain a little bit?**

Thank you for highlighting this important point. We believe that discussing these terms makes the manuscript more strengthen. We mentioned them in the Introduction section of the revised manuscript, please see Page 2. We also changed the organization in the revised manuscript and mention the evolution of blockchain technology in the healthcare management field. We discuss data commons and data ecosystems with blockchain timeline too. We believe that blockchain has a potential for contributing to data ecosystems.

**(4) The Figure 5 is hollow, and it is a little bit hard for the readers to differentiate the colors. I would recommend making it solid, with colors filled in.**

Authors fully agree with the intent of this comment and all figures are edited in the revised manuscript. Also, we would like to mention that some of the figures are totally changed in the revised manuscript, to address some other comments, e.g., removing stopped projects from the paper.

## Reviewer 4

Thanks for all the suggestions. In the revised manuscript, all comments have been carefully addressed and have been answered below respectively.

**(1) Paper is informal in many paragraphs; the authors must have corrected it. English must have improvements**

We have proofread the paper and edited the text accordingly. We improved the English to ensure that an international audience can follow the text.

**(2) Recent resources must add to references, for example, 2020 and 2021 papers.**

Thank you very much for raising such an important point. In the revised manuscript, 9 new studies from 2019-2021 have been added. They are available in section 3.2 "Blockchain Development Era in EHR Sharing" Page 12-15 and section 3.3 "Blockchain As a Platform Era in EHR Sharing" Page 15-18. The suggested references (Ramachandran et al 2020, Polap et al. 2021 and Srivastava et al. 2020) are also examined in the revised manuscript. After analyzing these recent studies, we changed the overall structure in the revised manuscript. Please note that the timeline has been created and the studies are discussed based on the blockchain evaluation timeline. We believe that this change makes the manuscript much stronger in the revision. We thank the reviewer for this suggestion.

**(3) The introduction section is too short to detail the motivations and challenges of this paper.**

We thank the reviewer for this suggestion. In the revised manuscript, "Introduction" section, Page 2, is edited and detailed. Introduction section is divided into subheadings in order to explain the details and to address this comment.

**(4) Environmental features should be considered in terms of classification. Healthcare environment characteristics must have more detail in this paper; it is necessary to identify the main features of the health care areas, a taxonomy or a framework must have provided, and based on that, the effects of using the block chain to become apparent. It can help in choosing methods in other articles or upcoming developments**

We thank the reviewer for this suggestion. After analyzing recent studies, we changed the overall structure in the revised manuscript. Please note that the timeline has been created and the studies are discussed based on the blockchain evaluation timeline. We believe that this change makes the manuscript much stronger in the revision. We thank the reviewer for this suggestion. Also, data ecosystem and data commons terms are discussed in the revised manuscript. Please see the yellow highlighted parts in the revised manuscript.

**(5) It should be compared with previous review papers. And its superiority over previous works should be determined. It is suggested that new articles that have been reviewed in recent years be reviewed to determine the advantages of the article over them. This article should have a special advantage over other articles or previous reports that can be used later. For this reason, it is suggested that in some cases, the studies be done in more depth and detail and the author's analysis and conclusion be added to it.**

We thank the reviewer for this suggestion. In section 1.5 "Rationale of the Review and Intended Audience" Page 5, 1$^{st}$ Paragraph of the revised manuscript, we showed our contribution and advantages over existing review articles.

Also, we explained our differences from other articles in order. The section 4 "Discussion" Page 20, 3rd and 4th Paragraphs, and section 5 "Conclusions" sections Page 21, 2nd Paragraph of the revised manuscript have also been developed and edited in line with comments. The latest approaches in medical blockchains are examined in the revised manuscript and 9 new studies from 2019-2021 have been added. They are available in section 3.2 "Blockchain Development Era in EHR Sharing" Page 12-15 and section 3.3 "Blockchain As a Platform Era in EHR Sharing" Page 15-18. Section 3.3 "Blockchain As a Platform Era in EHR Sharing" Page 15-18, focused on studies that use AI methods in conjunction with medical blockchains, and details are given.

**(6) It is suggested that non-functional requirements are categorized, and the effects of using the block chain be illustrated.**

Blockchain is a distributed and decentralized system that works on a P2P network. Also, the smart contract mechanism is using its own virtual machine. When we build a proper design, it is possible to use software tools and/or apps on a blockchain platform. We can consider a blockchain application as a distributed app (Dapp). Recently, interoperability of different blockchain networks is also possible. Thus, it can be mentioned that blockchain-based platforms can support cloud computing or software tools distributedly. In the Introduction section of the revised manuscript, please see Page 2. We also changed the organization in the revised manuscript and mention the evolution of blockchain technology in the healthcare management field. We discuss data commons and data ecosystems with blockchain timeline too. We believe that blockchain has a potential for contributing to data ecosystems. Please also see our answer to the next question.

**(7) The authors must identify important parameters for evaluating each method or algorithm**

Authors fully agree with the intent of this comment. All studies have been carefully analyzed, but it is not possible to compare all systems in practice and theory. The reason is that the studies that emerged in the first years are only theoretical, and only a few of the latest studies have implemented blockchain-sided implementations. The studies conducted in recent years have used other methods, such as AI with blockchain, and evaluated system performance by these methods. We try to find the most suitable metrics that can be used for the comparisons. There is still not enough technical information presented in the articles to have statistics for the comparisons. We, finally, decide to show the blockchain evolution timeline to compare the similar type of projects within the group. We also would like to discuss non-functional requirements too, but it is also not possible based on the given technical information. It is out of scope of this article.

**(8) The simulation and implementation part of the article is examined and added to the tables, and converted into graphs if necessary**

We thank the reviewer for this suggestion. We provide figures and tables to compare the projects and articles. It is not possible to convert the collected data into graphs since we mention the common advantages/disadvantages. But, we provide a new figure in the revised manuscript to improve the presentation according to address this comment.

**(9) Examples given in the text should be transferred to health-related areas**

We thank the reviewer for this suggestion. In line with the comments, the explanation of the double-spending problem is edited in section 1.2 "General Structure of Blockchain Technology" Page 3, 1st and 2nd Paragraphs.